# The Importance of Antioxidant Activity for the Health-Promoting Effect of Lycopene

**DOI:** 10.3390/nu15173821

**Published:** 2023-08-31

**Authors:** Anna Kulawik, Judyta Cielecka-Piontek, Przemysław Zalewski

**Affiliations:** 1Department of Pharmacognosy and Biomaterials, Faculty of Pharmacy, Poznan University of Medical Sciences, 3 Rokietnicka St., 60-806 Poznań, Poland; anna.kulawik@student.ump.edu.pl (A.K.); jpiontek@ump.edu.pl (J.C.-P.); 2Phytopharm Klęka S.A., Klęka 1, 63-040 Nowe Miasto nad Wartą, Poland

**Keywords:** lycopene, neuroinflammation, neurodegenerative diseases

## Abstract

Lycopene is a compound of colored origin that shows strong antioxidant activity. The positive effect of lycopene is the result of its pleiotropic effect. The ability to neutralize free radicals via lycopene is one of the foundations of its pro-health effect, including the ability to inhibit the development of many civilization diseases. Therefore, this study focuses on the importance of the antioxidant effect of lycopene in inhibiting the development of diseases such as cardiovascular diseases, diseases within the nervous system, diabetes, liver diseases, and ulcerative colitis. According to the research mentioned, lycopene supplementation has significant promise for the treatment of illnesses marked by chronic inflammation and oxidative stress. However, the majority of the supporting data for lycopene′s health benefits comes from experimental research, whereas the evidence from clinical studies is both scarcer and less certain of any health benefits. Research on humans is still required to establish its effectiveness.

## 1. Lycopene

Lycopene is a compound of red and orange fruit and vegetables, like tomato, watermelon, papaya, pink guava, carrot, rosehip, apricot, pink grapefruit, and pumpkin [1,2,3]. It is responsible for their red and orange color due to light absorption with a maximum wavelength of λ = 444, 470, and 502 nm [2]. It takes part in the process of photosynthesis and protects plants from damage caused by overexposure to light. It is also an essential intermediate in the synthesis of beta-carotene and xanthophylls [4]. However, lycopene is also found in some plants that have other colors, such as parsley and asparagus [5].

The most common source of lycopene in diets are tomatoes and products containing tomatoes [1,2,3]. More than 85% of this ingredient in our diet comes from these sources [6]. Tomatoes are also the cheapest source for lycopene production [7]. The tomato-based products are a better source of this compound than raw tomatoes [1]. Different varieties of tomatoes, as well as other fruits and vegetables, contain different amounts of this ingredient [3].

The amount of this carotenoid is affected by various factors, such as the degree of maturity of the plant material, fruit variety, light, temperature, climate, irrigation, location of plantation, soil quality, processing, and conditions of storage [3,8]. When the temperature exceeds 35 °C, the amount of lycopene decreases because it is converted to beta-carotene. The lycopene content can increase by about 36% when grown in soil containing the necessary microorganisms [2,3]. The amount of this carotenoid is higher in ripe fruits because they contain less chlorophyll [8]. Light and oxygen have the greatest impact on storage and processing. Its synthesis in tomatoes is promoted via supplementation with red, far-red, and blue light [9,10]. The content of lycopene in tomato puree shows the highest stability (with a total loss of 20%). It has high stability in comparison with other substances, such as ascorbic acid, kaempferol, and quercetin, and also after multiple sterilization and evaporation cycles. Its stability may be related to the presence of, e.g., ascorbic acid and phenolic compounds in tomatoes, and also to the influence of these compounds on the inhibition of the process of isomerization and autooxidation of lycopene. They ensure greater stability compared to pure lycopene [8].

The human body cannot synthesize lycopene. It must be supplied with the diet [11]. Its intake varies by region of the world. In Europe, its consumption from natural sources is 0.5–5 mg/day. In the United States, its daily supply is over 7 mg/day. It is possible to consume 20 mg of this compound daily with a higher consumption of vegetables, fruits, and tomato-based products. Lycopene from natural sources accounts for 50 to 65% of the total intake of this ingredient [3]. Chewing and peristalsis are needed to increase the bioavailability of this carotenoid. These processes mechanically disturb the food. This enables the lycopene in the food matrix to be released [12,13]. In the stomach, it is subjected to the action of digestive enzymes, gastric acid, and the mechanical movements of the stomach. These factors will also contribute to its release from the food matrix. Lycopene can be incorporated into lipid droplets and pass into the small intestine. There, the next stage of decomposition of the food matrix takes place with the help of bile acids and enzymes [13]. Lycopene can be incorporated into lipid micelles and then absorbed by enterocytes [14,15]. Absorption of this compound can occur via passive diffusion. Its absorption may be mediated by the class B scavenger receptor type 1 (SR-B1), which is involved in the absorption of other compounds from the group of carotenoids, such as lutein and beta-carotene [14,16,17]. Due to the high expression of ß-carotene oxygenase 1 (BCO1) and ß-carotene oxygenase 2 (BCO_2_) in the intestine, partial cleavage of this carotenoid may occur in the enterocytes [18]. Most of the lycopene is incorporated into the chylomicrons unchanged and then passes into the lymphatic system [14,19]. The microsomal triglyceride transfer protein (MTTP) may be involved in this process. It is an enzyme that delivers lipids to the chylomicrons that are formed [20]. They pass from the lymph into the portal circulation. There, extrahepatic lipoprotein lipases break them down to chylomicron residues. The liver removes remnants of chylomicrons from the portal system. Lycopene is incorporated into very low-density lipoproteins that are carried into the bloodstream [21]. Many lifestyle and biological factors affect the absorption of this carotenoid. These are factors such as age, gender, blood lipid concentrations, hormonal status, body mass index and composition, the content of other carotenoids in food, and smoking and alcohol consumption [22]. Its bioavailability decreases with age and in the case of certain diseases [23]. Lycopene is a lipophilic compound, so the presence of fat in food enhances its absorption [24]. Its absorption is hampered by dietary fiber and beta-carotene [25]. After absorption from food, it is mainly stored in the liver, prostate, and adrenals. It is also present in lower concentrations, e.g., in the brain and skin [3,26]. The adipose tissue contains more lycopene than serum because it is a lipophilic compound [25]. The average serum concentration is 0.2–1.0 µmol/L [27]. The half-life of lycopene in human plasma is 12–33 days [1]. Its metabolism occurs in the liver [16]. Oxidative and enzymatic degradation leads to the formation of its metabolites. Biologically active metabolites include apo-lycopenals, apo-lycopenones, apo-carotenedials, epoxides, and carboxylic acids [28,29]. The main metabolites of lycopene are lycopene 1,2-epoxide and lycopene 5,6-epoxide, along with other minor metabolites like lycopene 1,2;5,6-diepoxide, lycopene 1,2;50,60-diepoxide, lycopene 5,6;50,60-diepoxide, and lycopene 1,2;10,20-diepoxide (Figure 1) [30].

Lycopene is classified as a carotenoid [31]. It is not classified as a precursor of vitamin A, due to the lack of the final b-ion ring, which is present in the structure of the vitamin A core. The molecular formula of lycopene is C_40_H_56_ [32]. Its molecular weight is 536.85 Da [33]. Chemically, this molecule is a linear hydrocarbon with 2 non-conjugated and 11 conjugated double bonds [34]. These bonds may undergo isomerization under the influence of factors such as temperature, light, and chemical reactions [2,35]. The isomerization leads to structures such as 5-cis, 9-cis, 13-cis, and 15-cis (Figure 2) [2,3,36]. With so many double bonds, it can theoretically exist in 1056 cis-trans configurations [37]. However, there are 72 isoforms found in nature [38]. Lycopene is sensitive to oxygen, light, heat, acids, metal ions, and catalysts [33,37]. The most stable form is 5-cis lycopene. The stability of the other isomers decreases in the following order: all-trans, 9-cis, 13-cis, 15-cis, 7-cis, and 11-cis [37]. Its isomerization affects its activity and bioavailability [39]. Lycopene isomers differ in their physical and chemical properties. They have a different melting point, polarization, color intensity, solubility, and ability to crystallize [40].

Lycopene of natural origin occurs as trans isomers [41]. However, in the human body, it exists as *cis* isomers. Isomerization conversion occurs during the storage, processing, and transport of food products and during metabolic processes in the human body [3]. The heat treatment process of food increases the bioavailability of lycopene. This is related to the transformation of its trans form into the cis form [42]. These processes also help to release this carotenoid from the plant matrix, which also improves its absorption [24]. The cis isomers have greater solubility in bile acids and are more readily absorbed in the colon [42,43]. They are also better absorbed into the bloodstream due to their smaller crystal sizes [44].

The acyclic structure of lycopene affects its solubility [2]. It shows good solubility in organic solvents like acetone, chloroform, benzene, hexane, methylene chloride, and petroleum ether [2,33]. It also has good solubility in carbon disulfide [33]. It is slightly soluble in ethanol and methanol [2]. However, it is insoluble in water [45].

Lycopene in the aqueous environment accumulates and precipitates in the form of crystals. It occurs in the chromoplasts of tomatoes in the form of elongated needle-shaped crystals. It is also found deep within the polysaccharides membrane structure [40].

Lycopene has a wide biological activity. Many studies show that this carotenoid and tomato-based products containing lycopene protect work against many chronic diseases and alleviate their effects [46]. Studies show its positive effect on cardiovascular diseases, including the regulation of blood lipid levels [8,23,47,48,49]. Lycopene also showed antidiabetic properties [37,50]. Research proves its beneficial effects in nervous system disorders, including neurodegenerative diseases such as Parkinson′s disease and Alzheimer′s disease [51,52,53,54]. The positive effect of lycopene has also been observed in liver diseases and ulcerative colitis [16,55].

This review is intended to present the results of research on the effects of lycopene in these research areas. Many diseases are associated with oxidative stress and chronic inflammation. Researchers are interested in the health-promoting properties of lycopene as a compound that counteracts these conditions.

## 2. Antioxidant Effects of Lycopene

Oxidative stress is characterized by an imbalance between the amount of reactive oxygen species (ROS) produced and the amount eliminated by antioxidants [56]. Endogenous reactive oxygen and nitrogen species (RONS) are produced in the physiological state by enzymes, including nitric oxide synthase (NOS) and NADPH oxidase. They are a by-product of oxidation reactions catalyzed via metals or mitochondrial electron transport chain processes. Free radical peroxide (O_2_•−) is the primary precursor to other reactive oxygen and nitrogen species. It can react with other particles to generate other free radicals such as hydroxyl (OH•), peroxyl (ROO−), and alkoxy (RO−), as well as H_2_O_2_. During the reaction of nitric oxide with free radical peroxide, the free radical peroxynitrite (ONOO−) is formed [57]. Antioxidant defense systems control reactions that produce reactive oxygen and nitrogen species. It deactivates free radicals and molecules that can turn into RONS [58]. Enzymes such as glutathione peroxidase (GSH-Px), catalase (CAT), and superoxide dismutase (SOD) are essential to this process [59]. Other endogenous antioxidants are also glutathione (GSH), melatonin, lipoic acid, uric acid, and bilirubin [60]. Oxidative stress contributes to cell and tissue damage via the multiple pathways. It is one of the main causes of chronic diseases. These include atherosclerosis, liver diseases, ulcerative colitis, and cardiovascular and nervous system disorders, including neurodegenerative disorders [61,62,63,64,65,66,67,68].

Lycopene is a molecule that most effectively quenches singlet oxygen from carotenoids [1]. Lycopene quenches singlet oxygen twice as much as beta-carotene and 10 times as α-tocopherol [69]. These properties result from its chemical structure-especially the system of conjugated double bonds. And to a lesser extent, it is affected by cyclic or acyclic end groups [1]. Among all the lycopene isomers, the greatest antioxidant properties are shown by the 5-cis form, followed by 9-cis, 7-cis, 13-cis, 11-cis and all-trans [37]. This is probably related to the better solubility and lower self-aggregation in the polar environment by cis forms [28]. The antioxidant activity of lycopene is greater in the case of tomato extracts than in the case of pure lycopene. This is due to its synergistic effect with other compounds such as beta-carotene, phytofluene, and phytoene [70]. Lycopene has the ability to reduce reactive oxygen species (ROS) and eliminate singlet oxygen, nitrogen dioxide, hydroxyl radicals, and hydrogen peroxide [70,71]. Its effect on reactive oxygen species includes radical attachment, electron transfer, and allylic hydrogen abstraction [70]. Lycopene can react with free radicals in more than one way [70,72]. The mechanisms of the antioxidant action of lycopene is presented in Figure 3 [37]. Many factors influence the reactivity of this compound in biological systems. These include the physical and molecular structures, concentration of lycopene, possibility of interaction with other antioxidants, partial pressure of oxygen, and place of action in the cell [73]. Adduct formation and allylic abstraction of hydrogen mainly occur in a non-polar environment. Electron transfer occurs in a polar medium [70]. Lycopene can act as a superoxide radical scavenger (LOO•) and as a singlet oxygen (1O_2_) [74]. It increases the levels of enzymatic antioxidants such as catalase, superoxide dismutase, and glutathione peroxidase. This is due to the activation of the antioxidant response element, which is associated with nuclear factor E2 (NFE2L2) [75]. Lycopene also has the ability to regenerate non-enzymatic antioxidants such as vitamin C and E. This has a positive effect on the cellular antioxidant defense system [71]. Thanks to its antioxidant properties, lycopene has the ability to protect structures important for the body, such as DNA and lipids [76].

The studies showed the antioxidant activity of lycopene. Pataro et al. [77] tested its activity using the FRAP (ferric reducing antioxidant power) method. They used lycopene extracted from tomato peels with ethyl lactate and acetone. The pulsed electric fields pre-treatment increased the antioxidant activity of all-trans form [77]. Amorim et al. [78] showed in the ORAC (oxygen radical absorbance capacity) test that there are more antioxidants in red guava than in tomatoes. Extracts from these plants were characterized by higher antioxidant activity than pure lycopene. This may be due to the content of other compounds with antioxidant potential in the extract. Lycopene and plant extracts’ Trolox equivalent antioxidant capacity (TEAC) values showed good correlations via simple linear regression analyses [78]. Stinco et al. [79], based on TEAC values, showed that lycopene may have a good ability to capture ABTS•+ (2,2′-azino-bis(3-ethylbenzothiazoline-6-sulfonic acid)) [79]. Alvi et al. [80] showed in the DPPH (2,2-diphenyl-1-picrylhydrazyl) test that this carotenoid has a high free radical scavenging activity. It was IC50 = 4.57 ± 0.23 μg/mL, and for vitamin C, IC50 = 9.82 ± 0.42 μg/mL. [80]. Wang et al. [81] compared the antioxidant activity of lycopene samples with different contents of Z isomers (5%, 30% and 55% of Z isomers). DPPH and ABTS tests showed that the antioxidant activity of lycopene increases with the content of Z isomers. In the DDPH test, for a sample containing 55% Z isomers, the IC50 value was 80 μg/mL; for a sample containing 30% Z isomers, it was 110 μg/mL; and for a sample containing 5% Z isomers, it was 140 μg/mL. In the ABTS assay, the IC50 values for the samples with 55%, 30%, and 5% Z isomers were 35, 60, and 80 μg/mL, respectively [81].

The studies evaluated the antioxidant activity of lycopene in different models and diseases [57].

## 3. Antioxidant Activity of Lycopene in Cardiovascular Diseases

Inflammation is a risk factor for cardiovascular disease. Chronic inflammation contributes to oxidative stress [82]. Increased oxidative stress is one of the key factors in cardiovascular disease. It also contributes to reperfusion, myocardial infarction, and heart failure [83]. Excess ROS affects the lower availability of nitric oxide and vasoconstriction, causing hypertension. ROS also negatively affect myocardial calcium levels, contributing to arrhythmias and increasing cardiac remodeling [84]. Oxidative stress can also lead to myocardial hypertrophy [85]. The generation of ROS under pathological conditions, like hypertension, surpasses the natural antioxidant capability. It can result in cell death. Myocardial infarction, reperfusion, and heart failure are all closely correlated with oxidative stress [83]. Another chronic disease associated with increased oxidative stress is atherosclerosis. Its pathogenesis is also associated with the activation of pro-inflammatory pathways and the expression of pro-inflammatory cytokines. In the course of this disease, there is an accumulation of lipids and inflammatory cells in the walls of the arteries [86].

Due to the role of oxidative stress in cardiovascular diseases, the use of antioxidants may bring health benefits [83]. Scientific evidence confirms that lycopene plays a beneficial role in the prevention of cardiovascular disease [49,87].

Shen et al. [88] studied the effect of lycopene on oxidative stress induced by di-(2-ethylhexyl) phthalate in the heart of SPF-grade ICR mice. Researchers observed that the tested carotenoid caused an increase in cardiac GSH-Px activity and an increase in cardiac GSH levels. There was also a decrease in the level of cardiac myeloperoxidase (MPO), cardiac H_2_O_2_, and a decrease in cardiac glutathione S transferase (GSH-ST) activity. The study showed that lycopene can inhibit oxidative stress caused by di-(2-ethylhexyl) phthalate [88]. Ferreira-Santos et al. [89] proved that a diet enriched with lycopene prevents hypertension caused by angiotensin II. It also helps to improve the remodeling of the cardiovascular system. Lycopene had an antioxidant effect by increasing the activity of GSH-Px and SOD in the liver [89]. In another study, researchers tested lycopene in myocardial inflammation caused by palmitate in Wistar rats. The tested carotenoid had a beneficial effect on the lipid profile and reduced oxidative stress. It lowered cardiac MDA and cardiac H_2_O_2_ levels and cardiac MPO activity and increased cardiac SOD activity and cardiac GSH levels. The tested carotenoid had an anti-inflammatory effect by reducing the expression of NF-κB mRNA in the heart. It decreased the level of IL-1β and IL-6 and increased the level of anti-inflammatory IL-10 in the heart [90]. Zeng et al. [85] showed in vitro (neonatal cardiomyocytes) and in vivo (C57/BL6J mice) studies that lycopene can inhibit myocardial hypertrophy by reducing oxidative stress. Reactive oxygen species generation increased during the hypertrophy process; however, lycopene reversed this trend and inhibited the activation of the ROS-dependent pro-hypertrophic mitogen-activated protein kinase (MAPK) and protein kinase B (Akt) signaling pathways. The tested carotenoid also activated the expression of antioxidant genes induced by the antioxidant response element [85]. He et al. [91] studied the activity of lycopene on a mouse model representing the state after myocardial infarction. The model was created via ligation of the left anterior descending coronary artery. Researchers observed a decrease in the expression of IL-1β and TNF-α and inhibition of the NF-κB pathway in the ischemic myocardium. The study proved that lycopene reduces inflammation and apoptosis of cardiomyocytes after a heart attack [91]. Another study proved the protective effect of lycopene in atrazine-induced Kunming mice heart inflammation. Researchers observed a decrease in the levels of pro-inflammatory mediators in heart: COX-2, TNF-α, IL-6, and IL-1β and an increase in the anti-inflammatory cardiac TGF-β1. Serum TNF-α levels were also lowered. Lycopene blocked the activation of the TRAF6-NF-κB pathway. It also reduced NO levels in heart and cardiac NOS activity [92].

Albrahim [93] studied the effects of lycopene on diet-induced hypercholesterolemia in Wistar rats. He observed an increase in the level of cardiac and hepatic SOD, CAT, GSH, GPx, and glutathione reductase (GR). In the heart and kidney, the tested carotenoid increased the level of GSH and decreased in the level of MDA and NO. Lycopene showed antioxidant activity. He also observed a decrease in the levels of cardiac and renal ICAM-1, TNF-α, IL-6, and NF-κB. The studied carotenoid showed an anti-inflammatory effect [93]. Alvi et al. [94] studied the role of lycopene in Sprague-Dawley rats with LPS-induced hypertriglyceridemia. The tested carotenoid regulated the hepatic gene expression of inflammatory markers: IL-1β, IL-6, TNF-α, and C reactive protein (CRP). It reduced their level in plasma. It reduced oxidative stress. This was measured using the FRAP test. It also increased the activity of paraoxonase-1 (PON-1) in plasma. Lycopene inhibited hepatocyte nuclear factor-1α, improved the sterol regulatory element-binding protein-2, and improved the LDL receptor. As a result, it reduced the expression of convertase subtilisin/kexin type-9 proproteins. It also decreased the affinity of Apo-CIII to bind to lipoprotein lipase [94]. In another study, red guava-derived lycopene reduced plasma triglyceride and plasma MDA levels and MPO activity in plasma in hypercholesterolemic diet-induced dyslipidemia hamsters. It also reduced hepatic steatosis. The study showed that lycopene has anti-atherosclerotic and hypolipidemic effects [95]. Colman-Martinez et al. [96] studied the effect of carotenoids (including lycopene) from tomato juice on markers of inflammation. It was an open, prospective, randomized, cross-over, and controlled clinical trial in people at high cardiovascular risk. Lycopene decreased the level of intercellular adhesion molecule 1 (ICAM-1) and vascular cell adhesion molecule 1 (VCAM-1) in plasma. The study showed that it can reduce the level of pro-inflammatory molecules that are linked to atherosclerosis [96]. Yang et al. [97] found in a study on the human umbilical vein cell line, EA.hy926, that lycopene inhibits ICAM-1 expression and inhibits the adhesion of TNFα-stimulated monocytes to endothelial cells. This effect is related to blocking the degradation of the IκBα inhibitory protein. Inhibition of the adhesion molecule expression was associated with blocking NF-κB activation. Lycopene also showed antioxidant activity. It increased the level of GSH and the expression of glutamate–cysteine ligase. It also activated nuclear factor-erythroid 2 related factor 2 (Nrf2). This affected the downstream expression of HO-1 [97].

Numerous in vitro and animal studies indicate that lycopene may be potentially beneficial in the treatment of cardiovascular disease, due to its antioxidant effects as well as other mechanisms, including its ability to inhibit inflammatory reactions. Unfortunately, there are few clinical studies on lycopene in the literature. This review included only one clinical trial in people at high cardiovascular risk. It has shown positive results in terms of preventing atherosclerosis [96]. In the future, this study should be the starting point for further human studies on the effects of lycopene in the treatment of atherosclerosis and other cardiovascular diseases.

## 4. Antioxidant Activity of Lycopene in Liver Diseases

One of the liver diseases is non-alcoholic fatty liver disease (NAFLD). It is a term that covers diseases from simple steatosis to functional and biochemical abnormalities in the liver. These can result in non-alcoholic steatohepatitis (NASH), which, in the further stage, may lead to cirrhosis and hepatocellular carcinoma [98]. Many factors contribute to NAFLD, including: oxidative stress, inflammation, insulin resistance, environmental factors, as well as genetic and epigenetic mechanisms [99]. Studies have shown that a key factor in the onset and development of NAFLD is chronic inflammation [100]. In the course of this liver disease, pro-inflammatory factors such as IL-1β, IL-6, and TNF-α are produced [101]. Research also indicates that the pyrin domain of the NOD-like receptor family containing three (NLRP3) inflammasomes plays an important role in the development of chronic inflammation and the progression of NAFLD [102]. In the course of NAFLD, there is an activation of NF-κB and an activation of the NLRP3 inflammasome pathway, which contributes to the maturation of caspase-1 and the production of pro-inflammatory IL-1β and IL-18 [103].

Oxidative stress has a significant role in the etiology of liver diseases. Reactive oxygen species set off an oxidative stress cascade, which damages mitochondria, resulting in lipid peroxidation, damaged cells and membranes, and ultimately leads to cell death [104].

Chang et al. [105] studied the effect of lycopene on liver damage in SPF-grade ICR mice caused by TiO_2_ nanoparticles. The studied carotenoid showed antioxidant properties. Researchers observed an increase in the liver in TAC and GSH levels and an increase in GSH-Px and SOD activity. They also observed a decrease in liver MDA levels [105]. Bandeira et al. [106], in a study on C57BL/6 mice with liver damage caused by acetaminophen, showed the antioxidant effect of lycopene. It reduced hepatotoxicity. It reduced the concentration of glutathione disulfide (GSSH) and increased the level of tGSH and CAT in the liver. It also reduced damage caused by oxidative stress by inhibiting protein carbonylation. It also reduced the activity of matrix metalloproteinase-2 (MMP-2). In an in vitro study on cells, SK-Hep-1 reduced ROS levels. This was due to NADPH inhibition in the protein kinase C pathway [106]. In another study, researchers showed that lycopene prevented non-alcoholic steatohepatitis in C57BL/6J mice. It inhibits and reverses insulin resistance caused by lipotoxicity. It weakens the accumulation of lipids in the liver and increases lipolysis. This is associated with a reduced recruitment of macrophages and T cells in the liver, as well as a reduction in the number of M1 macrophages/Kupffer cells. The beneficial effect of lycopene on the liver was related to its antioxidant activity. Lycopene reduced the mRNA levels of the M1 marker (TNFα, MCP1, and RANTES) induced by TNFα, LPS, and IFN-γ in peritoneal macrophages. The tested carotenoid also reduced the expression of fibrogenic genes in the stellate cell line caused by the transforming growth factor TGF-β1. In both cases, the effect was dose-dependent. The effect of the studied carotenoid was related to its influence on the expression of NADPH oxidase subunits, and thus, its antioxidant activity. The study showed that lycopene was anti-inflammatory [107]. Another study showed a beneficial effect of lycopene on non-alcoholic fatty liver disease in Sprague-Dawley rats. The tested carotenoid increased the activity of SOD and GSH-Px in the liver. It reduced liver histopathology and reduced liver weight. It also lowered LDL and total cholesterol [108]. Róvero Costa et al. [109] found that lycopene attenuated NAFLD in obese Wistar rats. It increased HDL levels and also decreased triglycerides and total cholesterol. It had an anti-inflammatory effect and reduced damage caused by oxidative stress. It reduced the levels of TNF-α and MDA. It also increased the total antioxidant capacity (TAP) as well as SOD and CAT activity in the liver [109]. Gao et al. [110] studied the effect of lycopene on the hepatic NF-κB/NLRP3 inflammasome pathway in C57BL/6J mice with NAFLD. The tested carotenoid reduced body weight gain and reduced white adipose tissue mass in mice. Mice receiving lycopene had lower levels of HDL, LDL, triglycerides, and total serum cholesterol. Lower serum insulin and glucose levels were also observed in these animals. These effects were dependent on the dose of the carotenoid studied. Lycopene affected the inflammatory signaling pathway associated with hepatic NLRP3 by reducing the expression of the liver proteins Pro-Caspase-1, Caspase-1, NLRP3, and NF-κB. The study showed that lycopene may prevent NAFLD by inhibiting the NF-κB/NLRP3 inflammasome pathway in the liver [110]. In another study, lycopene, luteolin, and their combination were shown to activate the adenosine monophosphate-activated protein kinase (AMPK) pathway. This reduces the "two hits" in NAFLD. The study was performed on sodium palmitate-induced fatty HepG2 cells and primary hepatocytes and on obese C57BL/6J mice fed a high-fat diet. Lycopene lowered the level of total cholesterol and triglycerides in the blood. It also lowered ALT and AST levels. Lycopene increased the expression of nicotinamide phosphoribosyl transferase (NAMPT). This contributed to the increase in the level of NAD+, a co-substrate of SIRT1, which influenced the inhibition of lipogenesis and increased the process of β-oxidation. Lycopene was also anti-inflammatory by inhibiting NF-κB activation and reducing IL-1β, IL-6, and TNF-α levels [111]. Mustra Rakić et al. [112] studied the effect of lycopene on NASH caused by cigarette smoking in ferrets. The tested carotenoid attenuated NASH by suppressing BCO2. It reduced inflammation in the liver. It had an antioxidant effect by increasing the expression of CAT, SOD2, and GSH-Px1 [112].

According to a number of experimental investigations, lycopene may be potentially beneficial in the treatment of liver problems because of its pleiotropic effect. The positive effect of lycopene is mainly related to its ability to suppress inflammatory responses as well as other mechanisms, including its antioxidant capabilities. To determine whether lycopene is a serious possibility for treating liver problems, however, further human trials are required.

## 5. Antioxidant Activity of Lycopene in Ulcerative Colitis

Ulcerative colitis is one of the subtypes of inflammatory bowel disease. It is characterized by chronic, recurrent inflammation [113]. The encouragement of excessive production of pro-inflammatory cytokines via ROS/RNS underlies the development of inflammation in inflammatory bowel disorders. This leads to a higher production of ROS/RNS and an intensification of oxidative/nitrosative stress [114].

Yin et al. [115] studied the effect of lycopene on the inflammation of the colon in ulcerative colitis caused by ochratoxin A in Sprague-Dawley rats. Researchers observed decreased levels of NO, MPO, MDA, and hydroxyproline and decreased IL-6 expression. They also observed an increase in GSH and SOD levels. The study showed that lycopene, thanks to its antioxidant properties, reduces oxidative stress and inflammation [115]. Kashef et al. [55] studied the effect of lycopene on the alleviation of acetic acid-induced ulcerative colitis in albino rats. The researchers observed an increase in GSH levels and a decrease in the average percentage area of COX-2 immunoexpression [55]. Baykalir et al. [116] studied the activity of lycopene in a rat model of acetic acid-induced colitis. The tested compound caused a decrease in the level of MDA and sialic acid. It also reduced fragmentation of DNA. Instead, it increased the levels of SOD, TAS, CPN, and iron [116].

Lycopene may be potentially helpful in the treatment of ulcerative colitis due to its capacity to reduce inflammatory responses as well as other mechanisms, including its antioxidant properties, according to a number of experimental studies. More human studies are needed, nevertheless, to assess whether lycopene is a serious candidate for treating ulcerative colitis.

## 6. Antioxidant Activity of Lycopene in Nervous System Disorders

Pro-inflammatory cytokines like TNF-α and IL-1β cause brain cell damage, and later, neuronal death. Inflammation is linked to the presence of oxidative stress. A pathogenic cascade that includes the activation of NF-κB and the production of other inflammatory mediators is set off by an excess of reactive oxygen species [117].

The nervous system, including the brain, is susceptible to oxidative damage. This is due to the fact that it requires a lot of oxygen; high demand to energy; high levels of iron, which is a strong catalyst for reactive oxygen species; the presence of many unsaturated lipids; and it is also associated with a relative deficiency in antioxidant defense. Oxidative stress is believed to be the main cause of brain aging [117,118,119].

Oxidative lipid damage plays an important role in the development of neurodegenerative diseases. This is due to the large amount of polyunsaturated fatty acids that occur in the lipid layer of the brain. An increase in the end products of lipid peroxidation is observed in Parkinson′s and Alzheimer′s diseases [120]. In the course of neurodegenerative diseases, abnormal protein aggregation occurs. In Alzheimer′s disease, β-amyloid and hyperphosphorylated tau proteins are accumulated. In Parkinson′s disease, α-synuclein accumulates. This protein leads to the formation of intracellular Lewy bodies. The aggregation of these proteins causes mitochondrial dysfunction, the generation of reactive oxygen species, microglial overactivation, and chronic inflammation, leading to DNA damage and apoptosis neurons in the brain [120,121].

Natural compounds with neuroprotective properties are currently being sought. This is due to the unavailability of therapies that slow or stop the progression of the disease. Studies have shown the potential of carotenoids, including lycopene, as neuroprotective agents [122]. Lycopene is a lipophilic compound, which makes it easier to penetrate the blood–brain barrier. This may explain why it has the ability to prevent or treat a variety of neurological illnesses [123].

Zhao et al. [117] studied the effect of lycopene on cognitive defects caused by oxidative stress. They conducted the study in CD-1 mice with D-galactose-induced neuroinflammation. The studied carotenoid corrected the amounts of brain-derived neurotrophic factor and favored the repair of histopathological damage. It increased the activity of antioxidant enzymes NAD(P)H quinone dehydrogenase 1 (NQO-1) and heme oxygenase 1 (HO-1). It lowered the level of IL-1β and TNF-α. It also increased the Iba-1 and GFAP expression. Lycopene reduced neuronal damage by activating Nrf2 and inactivating NF-κB translocations in a study on SH-SY5Y cells treated with H_2_O_2_ [117]. Wang et al. [124] studied the ability of lycopene to alleviate brain damage caused by sulfamethoxazole in grass carp. Exposure to sulfamethoxazole led to an increase in SOD activity, MDA, and 8-hydroxy-2 deoxyguanosine (8-OHdG) contents and a decrease in the content of GSH. Administration of lycopene caused these values to return to normal levels. The use of this carotenoid restored the activity of acetylcholinesterase. The use of this carotenoid decreased the expression of IL-1β, IL-6, IL-8, and TNF-α. Lycopene had also a neuroprotective effect by restoring the balance of the NF-κB/Nrf2 pathway. It had neuroprotective, anti-inflammatory, and antioxidant effects [124]. Zhang et al. [125], in a study on mice, showed that in lipopolysaccharide-induced neuroinflammation, lycopene reduced the level of IL-6 and TNF-α, which had a protective effect on brain tissue [125]. In another study, lycopene attenuated early brain damage as well as anti-inflammation after subarachnoid hemorrhage in Sprague-Dawley rats. Researchers observed an ameliorating effect on neurological deficits, neuronal apoptosis, cerebral edema, and the impairment of the blood–brain barrier. The tested carotenoid also showed anti-inflammatory effects by reducing the levels of TNF-α, IL-1β, and ICAM-1 [126].

Yu et al. [127] studied the effect of lycopene on oxidative stress and memory problems in tau transgenic mice expressing a P301L mutation. The tested compound decreased the concentration of MDA and increased the activity of GSH-Px. It also reduced tau hyperphosphorylation, which is associated with Alzheimer′s disease [127]. Lim et al. [128] studied the effects of lycopene in neuronal cells with the increased expression of RCAN1. It reduced intracellular and mitochondrial ROS levels and NF-κB activity. It also decreased mitochondrial respiration, Nucling expression, and glycolytic activity. Increased mitochondrial membrane potential was observed. Lycopene inhibited cell apoptosis, cytochrome c secretion, caspase-3 activation, and DNA fragmentation in the tested cells [128]. Qu et al. [129] studied the effect of lycopene on cultured rat cortical neurons treated with β-amyloid. It showed antioxidant activity. It reduced the production of peroxides in the mitochondria and the level of ROS. It also reduced the morphological changes in the mitochondria caused by β-amyloid. Lycopene improved the opening of mitochondrial transition pores and influenced the release of cytochrome c. It also increased the level of mitochondrial transcription factor A and protected mDNA against damage [129]. Wang et al. [130] studied the effects of lycopene on LPS-induced neuroinflammation and oxidative stress in C57BL/6J mice. The tested carotenoid prevented memory loss. It also prevented the accumulation of β-amyloid and decreased the amount of the amyloid precursor protein. It increased the expression of ADAM10 α-secretase and inhibited the neuronal BACE1 β-secretase. It decreased the expression of IBA-1, which is a marker of microglial activation. It had an antioxidant effect. It also showed anti-inflammatory effects by reducing the expression of iNOS, COX-2, and IL-1β and increasing the expression of the anti-inflammatory cytokine IL-10 in the hippocampus. It also showed antioxidant activity by increasing the level of GSH, SOD, and CAT activity [130]. Sachdeva et al. [131] studied the effect of lycopene on intracerebroventricular (ICV) Aβ1–42-induced neuroinflammation in a model of Alzheimer′s disease in Wistar rats. They observed a decrease in the activity of caspase-3 and the level of IL-1β, TNF-α, TGF-β, and NF-κB in rat brains [131]. Prema et al. [52] studied the effect of lycopene in C57bL/6 mice with Parkinson′s disease induced by 1-methyl-4-phenyl-1,2,3,6-tetrahydropyridine (MPTP). The tested carotenoid protected against a decrease in the level of dopamine and its metabolites in the striatum. It alleviated the occurrence of motor abnormalities. It also reduced oxidative stress by affecting the level of GSH and the activity of GPx, SOD, and CAT. The use of lycopene also affected the expression patterns of anti-apoptotic markers (cytochrome c and Bcl-2) and apoptotic markers (Bax, caspases-3, -8 and -9). Lycopene showed anti-apoptotic and antioxidant activity [52].

Table 1 summarizes the studies on the effects of lycopene on nervous system disorders that were reported in this review. These are in vitro studies and animal studies. In the literature data from recent years, there are no reports on clinical trials on the effect of lycopene on diseases of the nervous system, therefore they are not included in this review. The analyzed experimental study indicates the potential beneficial effect of lycopene in diseases of the nervous system, especially in neurodegenerative diseases, such as Alzheimer′s and Parkinson′s diseases. The mechanism of action of the studied carotenoid in the discussed diseases is complex. It is worth paying special attention to the antioxidant and anti-inflammatory properties of lycopene because chronic inflammation and oxidative stress occur at the base and in the course of many diseases of the nervous system. However, additional human studies are required to determine whether lycopene could be a potential drug for treating diseases of the nervous system.

## 7. Activity of Lycopene in Type 2 Diabetes Mellitus

Diabetes is a chronic disease and is one of the biggest public health problems in the world [132,133]. It is estimated that the number of people with diabetes in the world in 2021 was 536.6 million people, and in 2045, it will increase to 783.2 million people [134]. It is one of the top 10 causes of death worldwide [132,133]. There are three main types of diabetes: type 1, type 2, and gestational. About 90% of people with this disease have type 2 diabetes [133].

Type 2 diabetes is caused by insulin resistance, impaired insulin secretion, or both. Diabetes leads to changes in the metabolism of carbohydrates, proteins, and fats. The result of insulin resistance is high blood glucose levels caused by hindering the uptake and use of glucose by the body′s cells. As diabetes progresses, secondary pathophysiological changes develop in many organs, such as retinopathy, nephropathy, neuropathy, and cardiovascular diseases [135].

Diabetes causes hyperglycemia. It leads to glucose oxidation, oxidative degradation, and the non-enzymatic glycation of proteins. In diabetes, hyperglycemia is the main cause of increased levels of free radicals and reactive oxygen species. This leads to oxidative stress and lipid peroxidation and affects the level of antioxidant defense. This leads to further disorders of glucose metabolism. Oxidative stress contributes to the development of type 2 diabetes and related diseases [136].

Many studies confirm the connection between type 2 diabetes and oxidative stress. This relationship was determined by examining biomarkers of oxidative stress in patients with type 2 diabetes [137,138,139,140,141,142]. People with type 2 diabetes had lower glutathione peroxidase activity compared to people in the control group [137,139,140,142]. Jiffri et al. [139] and Mandal et al. [141] observed less superoxide dismutase activity. However, Aouacheri et al. [137] observed an increase in this parameter. Picu et al. [142] found that SOD activity did not change. Patients with type 2 diabetes had decreased levels of GSH and CAT [137,139,140,141,142]. Malondialdehyde, a marker of oxidative stress, was increased in people with this disease [137,138,139,140,141]. Mandal et al. [141] observed a lower total antioxidant capacity level. Hypoglycemia and oxidative stress can lead to a reduction in the expression of CAT, SOD and GSH-Px in pancreatic β cells. Long-term oxidative stress leads to an inhibition of insulin secretion by pancreatic β cells [143].

Chronic inflammation is one of the causes of oxidative stress in type 2 diabetes [144]. Increased formation of advanced glycation end products contributes to oxidative stress in metabolic disorders in diabetes [145]. Oxidative stress also stimulates glyco-oxidation reactions, during which advanced glycation end products are formed [146].

Studies showed increased levels of TNF-α, IL-6, and CRP in patients with type 2 diabetes [138,139].

Preventing and delaying the development of type 2 diabetes is connected with lifestyle modification or the use of some pharmacologically active substances [133]. Diet also plays an important role in the development and treatment of type 2 diabetes. It is essential in the prevention of this disease. Some food ingredients have significant health value [147]. Studies indicate that lycopene and tomato products have good prospects for the prevention and treatment of type 2 diabetes (Table 2) [136,147,148,149,150,151,152,153,154,155,156].

Lycopene reduces the risk of developing type 2 diabetes and has a beneficial effect in its therapy by regulating the various signaling pathways and the anti-inflammatory and antioxidant effects [157,158].

**Table 2 nutrients-15-03821-t002:** Effects of lycopene in diabetes mellitus.

Disease	Model/Participants/Type	Period	Lycopene Dosage and Administration	Main Results	Year Published	Reference
Diabetes mellitus, diabetic nephropathy	Streptozotocin-induced diabetic nephropathy male Kunming mice	8 weeks	Lycopene dissolved in the vehicle, 40 and 80 mg/kg, three times a week, intragastrically administration	1. Decreased FBG,	2015	[154]
				2. Increased the body weight,		
				3. Increased SOD and GSH-Px levels,		
				4. Decreased MDA level,		
				5. Reduced proteinuria,		
				6. Upregulated intrarenal HO-1 level,		
				7. Attenuated NF-κB and TNF-α expressions in kidney,		
				8. Increased HDL level,		
				9. Decreased LDL level		
Diabetes mellitus	Streptozotocin-induced diabetic Sprague-Dawley rats	30 days	Lycopene in sunflower oil, 10 mg/kg/day, oral gavage	1. Decreased blood and urine glucose levels,	2016	[136]
				2. Decreased vacuolization in the pancreas,		
				3. Increased serum insulin levels		
Diabetes mellitus	Streptozotocin-induced diabetic male Wistar rats	28 days	Lycopene in corn oil, 4 mg/kg/day, oral gavage	1. Decreased blood glucose levels,	2016	[150]
				2. Increased GSH-Px, SOD, GST, and CAT levels in liver,		
				3. Decreased MDA level in liver,		
				4. No significant changes in the levels of hemoglobin, RBC, hematocrit, MCHC, MCV, and MCH,		
				5. Mitigated histopathological changes in the liver,		
				6. Increased AST, LDH, ALT, and ALP activities,		
				7. No significant changes in TGL and total cholesterol levels		
Diabetes mellitus	Alloxan monohydrate-induced diabetic male and female Wistar rats	14 days	Lycopene niosomes, 100 and 200 mg/kg/day, lycopene extract 200 mg/kg/day, oral administration	1. Decreased blood glucose levels,	2017	[151]
				2. Decreased total cholesterol, TGL, LDL, and VLDL		
Diabetes mellitus	Streptozotocin-induced diabetic male and female Wistar rats	4 weeks	Lycopene in olive oil, 10, 20, and 40 mg/kg/day, oral administration	1. Decreased erythrocyte osmotic fragility,	2017	[152]
				2. Decreased erythrocyte MDA concentration		
Diabetes mellitus	Streptozotocin-induced diabetic male Wistar rats	50 days	Tomato extract mixed with plain yogurt (4.5 mg/kg/day lycopene), oral gavage	1. Decreased glycemia,	2017	[155]
				2. Decreased serum ox-LDL and liver TBARS,		
				3. Increased CAT and SOD levels,		
				4. Increased NPSH groups level,		
				5. No significant changes in GSH-Px level,		
				6. Decreased triacylglycerol and total-cholesterol,		
				7. Increased HDL		
Diabetes mellitus	Streptozotocin-induced diabetic female Wistar rats	28 days	Lycopene, 4 mg/kg/day, oral gavage	1. Increased CAT, SOD, GSH-Px, and GST activity,	2017	[149]
				2. Decreased MDA level		
Diabetes mellitus	Streptozotocin-induced diabetic male Wistar rats	8 weeks	Lycopene, 4 mg/kg/day, oral administration	1. Increased TAC level,	2018	[153]
				2. Decreased MDA level		
Diabetes mellitus	Streptozotocin-induced diabetic male Sprague Dawley rats	10 weeks	Lycopene oil solution, 10 and 20 mg/kg/day, oral administration	1. Decreased FBG,	2019	[147]
				2. Decreased lipid in blood and liver,		
				3. Decreased GHb level,		
				4. Decreased HOMA-IR,		
				5. Increased insulin level,		
				6. Increased GSH-Px and SOD levels in pancreas,		
				7. Decreased MDA level in pancreas		
Diabetes mellitus	Streptozotocin-induced diabetic male Sprague-Dawley rats	10 weeks	Lycopene, 5, 10, and 15 mg/kg/day, intragastric gavage	1. Decreased FBG,	2019	[148]
				2. Decreased HOMA-IR,		
				3. Increased FINS,		
				4. Decreased GHb, ox-LDL, and MDA levels,		
				5. Increased CAT, SOD, and GSH-Px activity,		
				6. Decreased CRP and TNF-α		
Diabetes mellitus, diabetic retinopathy	Diabetic retinopathy: 272 patients, diabetes mellitus without retinopathy: 190 patients, control: 285 patients, cross-sectional study	No data	Dietary intake	1. No significant association between HbA1c and lycopene	2017	[159]
Diabetes mellitus	Type II diabetes mellitus: 87 patients, control: 122 patients, case–control study	12 months	Dietary intake, 0.04 mg/kg/day	1. HbA1c and fasting plasma glucose levels decreased significantly with higher lycopene intake	2021	[156]

ALP: alkaline phosphatase; ALT: alanine aminotransferase; AST: aspartate aminotransferase; CAT: catalase; CRP: C reactive protein; FBG: fasting blood glucose; FINS: fasted blood insulin; GHb: glycosylated hemoglobin; GSH-Px: glutathione peroxidase; GST: glutathione-S-transferase; HbA1c: glycated hemoglobin; HDL: high-density lipoprotein; HO-1: heme oxygenase 1; HOMA-IR: homeostatic model assessment for insulin resistance; LDH: lactate dehydrogenase; LDL: low-density lipoprotein; MCH: mean corpuscular hemoglobin; MCHC: mean corpuscular hemoglobin concentration; MCV: mean corpuscular volume; MDA: malondialdehyde; NF-κB: nuclear factor-kappa B; NPSH: non-protein sulfhydryl; ox-LDL: oxidized low-density lipoprotein; RBC: red blood cell; SOD: superoxide dismutase; TAC: total antioxidant capacity; TBARS: thiobarbituric acid reactive substances; TNF-α: tumor necrosis factor alpha; TGL: triglycerides; VLDL: very-low-density lipoprotein.

Lycopene increases the antioxidant status and reduces damage caused by oxidative stress in type 2 diabetes. This is related to its antioxidant properties and the induction of endogenous antioxidant enzymes. Lycopene also helps to protect the body against metabolic complications related to the glycation process [157]. Studies show that lycopene increased the activity of SOD, GSH-Px, GST, CAT, and total antioxidant capacity in the case of diabetes [147,148,149,150,153,154,155]. Lycopene decreased the MDA level in animal diabetes studies [147,148,149,150,152,153,154].

Studies showed that lycopene had an anti-inflammatory effect in diabetes mellitus. It reduces the level of CRP and TNF-α [148,154].

Yin et al. [147], thought that lycopene′s ability to lower blood glucose levels and improve insulin resistance and lipid metabolism is a result of its antioxidant activity. Lycopene has the ability to capture free radicals and repair damage caused by oxidation. Thanks to this, it can activate pancreatic enzymes and reduce the process of glucose and lipid peroxidation [147]. Ozmen et al. [55] found that lycopene has a beneficial effect on pancreatic β-cells. It stimulates the secretion of insulin, which leads to a decrease in blood glucose [136]. Many other studies also confirm that lycopene lowers blood glucose levels [147,148,150,151,154,155].

According to the available research, lycopene′s antioxidant qualities help prevent and cure type 2 diabetes among other things by lowering the indicators of oxidative stress and triggering antioxidant defense mechanisms. The mechanism of action of lycopene in diabetes is complex and is not based solely on antioxidant activity. Lycopene also has a positive effect on pancreatic β-cells, stimulates insulin secretion, and has anti-inflammatory properties. Lycopene ingestion has been shown to benefit the prevention and management of type 2 diabetes in animal trials. The evidence for a protective association between lycopene intake and type 2 diabetes is currently limited in human research. Therefore, well-planned clinical trials are necessary to clarify and validate the role of lycopene in the treatment of diabetes.

## 8. Conclusions

Lycopene has a wide biological activity. Several in vitro and in vivo studies presented in this review provide valuable insights into the mechanism of action of lycopene and demonstrate its potential usefulness in conditions such as cardiovascular problems, including atherosclerosis, ulcerative colitis, and nervous system disorders including neurodegenerative diseases, diabetes, and liver disease including non-alcoholic fatty liver disease. The positive effect of lycopene is the result of its pleiotropic effect. Most of the evidence for lycopene′s beneficial effects on the aforementioned diseases comes from experimental work, while the evidence from clinical trials is both less abundant and less clear about any beneficial effect. The experimental studies cannot, however, be applied directly to people. Comparing research using animals is similarly challenging. Different dosages, various treatment periods, methods of delivery, as well as various animal species and their unique metabolic pathways all contribute to the differences in experimental design. Another limitation is the low oral bioavailability of this carotenoid, which limits its clinical usefulness. Nevertheless, the described studies suggest that lycopene supplementation has great potential in the treatment of diseases in which oxidative stress and chronic inflammation are observed. However, it is still necessary to perform research on people confirming its action.

## Figures and Tables

**Figure 1 nutrients-15-03821-f001:**
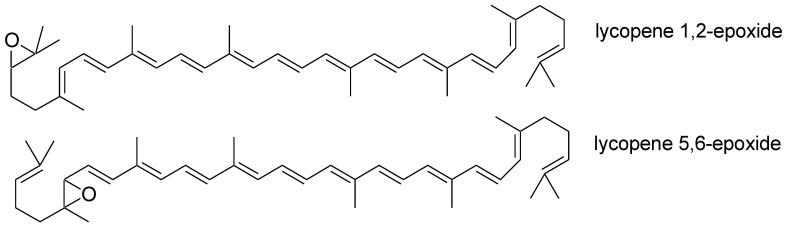
Chemical structure of the main lycopene metabolites.

**Figure 2 nutrients-15-03821-f002:**
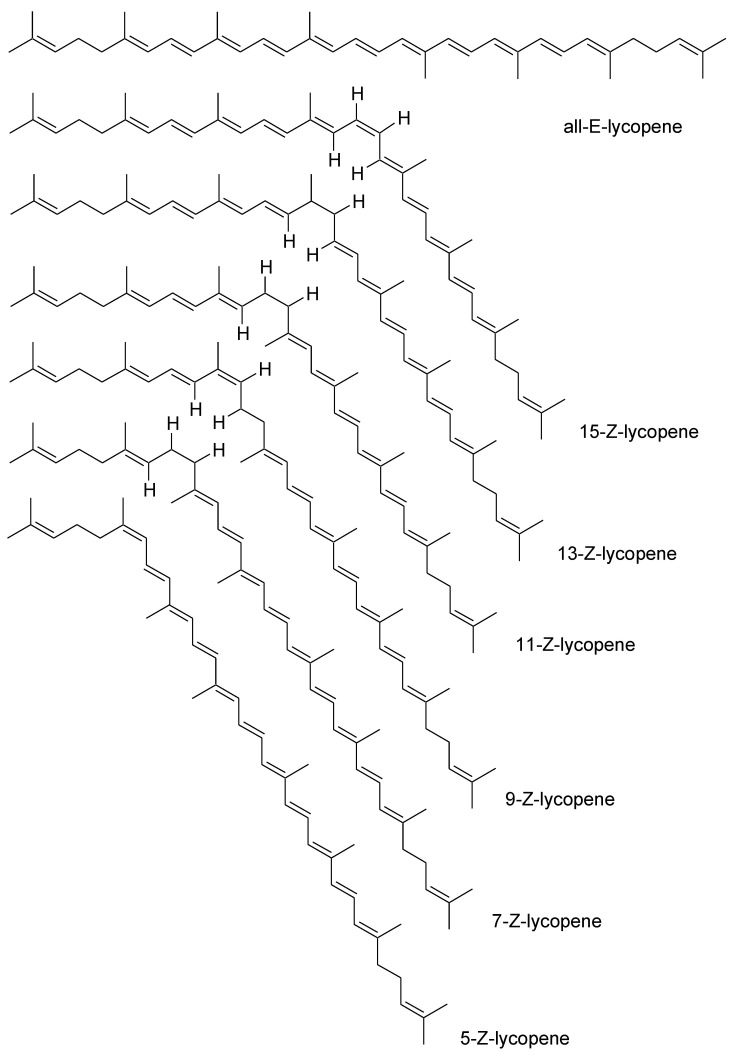
Various trans-cis isomeric forms of lycopene [36].

**Figure 3 nutrients-15-03821-f003:**
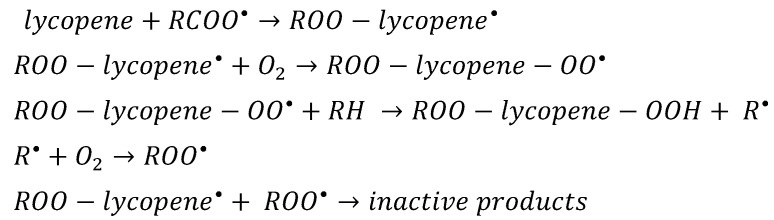
Mechanisms of lycopene′s antioxidant effect [37].

**Table 1 nutrients-15-03821-t001:** Effects of lycopene in nervous system disorders.

Disease	Model	Period	Lycopene Dosage and Administration	Main Results	Year Published	Reference
Neuroinflammation	CD-1 mice with D-galactose-induced neuroinflammation	8 weeks	50 mg/kg lycopene per day in diet	1. Corrected the amounts of brain-derived neurotrophic factor,	2017	[117]
				2. Favored the repair of histopathological damage,		
				3. Increased the activity of antioxidant enzymes NAD(P)H quinone dehydrogenase 1 (NQO-1),		
				4. increased the activity of heme oxygenase 1 (HO-1),		
				5. Lowered the level of IL-1β and TNF-α,		
				6. Increased the Iba-1 and GFAP expression		
Neuroinflammation	SH-SY5Y cells treated with H_2_O_2_	6 h	50 µM lycopene	1. Activated Nrf2 translocation,	2017	[117]
				2. Inactivated NF-κB translocation		
Neurotoxicity	Grass carps with sulfamethoxazole-induced neurotoxicity	30 days	basal diet supplemented with 10 mg/kg lycopene	1. Returned normal SOD activity, MDA, 8-hydroxy-2 deoxyguanosine (8-OHdG) levels, GSH content,	2022	[124]
				2. Restored acetylcholinesterase activity,		
				3. Decreased the expression of IL-1β, IL-6, IL-8, and TNF-α,		
				4. Restored the NF-κB/Nrf2 pathway balance		
Neuroinflammation	ICR male mice with lipopolysaccharide-induced neuroinflammation	7 days	60 mg/kg lycopene dissolved in 1% CMC as vehicle (carboxymethylcellulose sodium), oral gavage	1. Decreased expression of IL-1β and HO-1 in the hippocampus,	2016	[125]
				2. Decreased level of IL-6 and TNF-α in the plasma,		
				3. Relieved neuronal cell injury in hippocampal CA1 regions,		
				4. Pretreatment with lycopene ameliorated depression-like behaviors		
Early brain injury and inflammation following subarachnoid hemorrhage	Male Sprague-Dawley rats	1 day	40 mg/kg lycopene (dissolved in tetrahydrofuran), intraperitoneal injections	1. Ameliorated neurological deficits, neuronal apoptosis, cerebral edema, and impairment of the blood–brain barrier,	2015	[126]
				2. Decreased TNF-α, IL-1β, and ICAM-1 levels		
Alzheimer′s disease	Male P301L transgenic mice	8 weeks	5 mg/kg lycopene, oral gavage	1. Ameliorated the memory deficits,	2017	[127]
				2. Decreased MDA level,		
				3. Increased GSH-Px activity,		
				4. Attenuated tau hyperphosphorylation at multiple AD-related sites		
Alzheimer′s disease	Human neuroblastoma (SH-SY5Y) cells transfected with pcDNA containing HA-tagged human wild-type RCAN1	6 h	Lycopene dissolved in tetrahydrofuran, final concentration: 1 or 2 µM	1. Decreased intracellular and mitochondrial ROS levels,	2017	[128]
				2. Decreased NF-κB activity and Nucling expression,		
				3. Decreased in mitochondrial membrane potential, mitochondrial respiration, and glycolytic function in RCAN1-overexpressing cells,		
				4. Inhibited cell death, DNA fragmentation, caspase-3 activation, and cytochrome c release in RCAN1-overexpressing cells		
Alzheimer′s disease	Primary cultured Sprague-Dawley rat cortical neurons	24 h	Lycopene dissolved in tetrahydrofuran containing 0.025% butylated hydroxytoluene, added to culture medium, concentration of lycopene: 0.1, 0.5, 1, 2 or 5 µM	1. Decreased intracellular reactive oxygen species generation and mitochondria-derived superoxide production,	2016	[129]
				2. Ameliorated Ab-induced mitochondrial morphological alteration,		
				3. Opened of the mitochondrial permeability transition pores,		
				4. Released cytochrome c,		
				5. Improved mitochondrial complex activities,		
				6. Restored ATP levels,		
				7. Prevented mitochondrial DNA damages,		
				8. Improved the protein level of mitochondrial transcription factor A		
Alzheimer′s disease	Male C57BL/6J mice treated via intraperitoneal injection of LPS	5 weeks (35 days)	Lycopene (0.03%, *w*/*w*) mixed with standard diet	1. Prevented accumulation of Aβ,	2018	[130]
				2. Decreased levels of amyloid precursor protein,		
				3. Suppressed neuronal β-secretase BACE1,		
				4. Elevated the expressions of α-secretase ADAM10,		
				5. Downregulated the expression of IBA-1,		
				6. Reduced the expression of iNOS, COX-2, and IL-1β,		
				7. Increased the expression of the anti-inflammatory cytokine IL-10,		
				8. Increased the level of GSH, SOD, and CAT activity		
Alzheimer′s disease	BV2 microglial cells treated with LPS	8 h	Lycopene dissolved in dimethylsulfoxide concentration of lycopene: 12.5, 25, 50 µM	1. Suppressed the phosphorylation of MAPKs and NFκB,	2018	[130]
				2. Activated Nrf2 signaling pathways		
Alzheimer′s disease	Male Wistar rats	14 days	1 mg/kg, 2 mg/kg, and 4 mg/kg lycopene, oral gavage	1. Decreased activity of caspase-3,	2015	[131]
				2. Decreased the level of IL-1β, TNF-α, TGF-β, and NF-κB,		
				3. Remediated Aβ-induced learning and memory deficits,		
				4. Reduced Aβ1–42-induced mitochondrial dysfunction		
Parkinson′s disease	Male C57bL/6 mice injected intraperitoneally with MPTP	7 days	5, 10, and 20 mg/kg/day lycopene, orally	1. Protected against a decrease in the level of dopamine and its metabolites,	2015	[52]
				2. Attenuated MPTP-induced oxidative stress,		
				3. Attenuated motor abnormalities		
				4. Increased the level of GSH and the activity of GPx,		
				5. Decreased the activity of SOD and CAT		

ADAM10: a disintegrin and metalloproteinase 10; BACE1: beta-secretase 1; CAT: catalase; COX-2: cyclooxygenase-2; GPx: glutathione peroxidase; GSH: glutathione; IL-10: interleukin-10; IL-1β: interleukin-1 beta; iNOS: inducible nitric oxide synthase; MAPK: mitogen activated protein kinases; MDA: malondialdehyde; NF-κB: nuclear factor-kappa B; RCAN1: regulator of calcineurin 1; SOD: superoxide dismutase; TGF-β: transforming growth factor beta; TNF-α: tumor necrosis factor alpha.

## Data Availability

Not applicable.

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
