# Peer review of "The Importance of Antioxidant Activity for the Health-Promoting Effect of Lycopene"

_nutrients, 2023, doi:10.3390/nu15173821_

Round 1

Reviewer 1 Report

This review paper discussed the antioxidant effect of lycopene in vitro and in vivo, which contributes to the health functions of lycopene. Generally, this topic is interesting, while the organization of the manuscript is not well performed, existing overlapping like the categories of different diseases (e.g., 3. cardiovascular diseases v.s. 4. atherosclerosis; 7.  neurodegenerative diseases v.s. 8. nervous system disorders), which make confusions. Concerning the contribution of antioxidant effect of lycopene to its health benefits, authors are also suggested to be cautious not to exclude other mechanisms besides the antioxidant effect, since it might lead to misunderstanding. In addition, the manuscript is not quite readable without any figures. More importantly, the writing seems superficial with the accumulation of literature, but little discussion or perspectives were provided, leading to the lack of depth. 

Other comments include grammatical or typing errors and format errors like line 497, and the abstract is not acceptable to be so short without useful information being provided. In addition, human-based evidence/literature should be provided to further support the theme.

Language should be further polished.

Author Response

First of all, thank you for thoughtful and thorough review. We have revised the manuscript based on the comments of reviewers. We would like to kindly ask you to proceed with a review. Please find below a response to comments.

This review paper discussed the antioxidant effect of lycopene in vitro and in vivo, which contributes to the health functions of lycopene. Generally, this topic is interesting, while the organization of the manuscript is not well performed, existing overlapping like the categories of different diseases (e.g., 3. cardiovascular diseases v.s. 4. atherosclerosis; 7.  neurodegenerative diseases v.s. 8. nervous system disorders), which make confusions.

Thank you for pointing this out, we agree with the Reviewer’s viewpoint. We have made the following changes:

Chapters 3. cardiovascular diseases and 4. atherosclerosis have been have been reworded and merged into one chapter 3 on cardiovascular diseases.

Chapters 7. neurodegenerative diseases and 8. disorders of the nervous system have been reworded and merged into one chapter 6 on diseases of the nervous system.

Concerning the contribution of antioxidant effect of lycopene to its health benefits, authors are also suggested to be cautious not to exclude other mechanisms besides the antioxidant effect, since it might lead to misunderstanding.

Thank you for pointing this out, we agree with the Reviewer’s viewpoint. We have corrected the manuscript

 In addition, the manuscript is not quite readable without any figures.

Thank you for pointing this out, we agree with the Reviewer’s viewpoint. We have made the following changes:

Figures showing the main lycopene metabolites (Figure 1) and lycopene isomers (Figure 2) have been added.

 More importantly, the writing seems superficial with the accumulation of literature, but little discussion or perspectives were provided, leading to the lack of depth.

Thank you for pointing this out, we agree with the Reviewer’s viewpoint. We have corrected the manuscript.

Other comments include grammatical or typing errors and format errors like line 497, and the abstract is not acceptable to be so short without useful information being provided. In addition, human-based evidence/literature should be provided to further support the theme.

Thank you for pointing this out, we agree with the Reviewer’s viewpoint. We have corrected the manuscript.

In the abstract, conclusions and in the content of the publication, information was added that the positive effect of lycopene is the result of its pleiotropic effect and has many mechanisms of action.

For each of the chapters describing the effects of lycopene in diseases, a summary and little discussion or perspectives have been added.

The abstract has been corrected. More information has been added.

Lycopene is not registered as a drug, so there are not many studies on humans so far. Information that most of the evidence for lycopene comes from experimental studies has been added to the abstract and conclusions. There is a need to conduct clinical trials to confirm the beneficial effects of lycopene in the treatment of the described diseases in humans.

Fixed some grammatical or typing errors and format errors.

Thank you for pointing this out, we agree with the Reviewer’s viewpoint. We have corrected the manuscript.

Reviewer 2 Report

This paper discussed the importance of the antioxidant effect of lycopene in inhibiting the development of such diseases as cardiovascular diseases, liver diseases, and diseases within the nervous system. An important point is that the majority of evidence of a beneficial effect of lycopene on above-mentioned diseases comes from experimental work, while the evidence from clinical studies is both less abundant and less clear on any beneficial effect. This needs to be clearly underlined in the abstract and the conclusions. At the same time, there is only one table. On the other hand, sometimes it is really repetitive. Authors should also pay attention to the points outlined below, listed in order of appearance.

1.     Line 15, please check the sentence “diseases of the etiology of etiology

2.     “1. Lycopene”, graphic representation of lycopene, isomers, and metabolic processes in the human body are recommended.

3.     Line 156-159, the information are repetitious with above two paragraph.

4.     Line 245-297, where were these oxidative stress related markers and inflammatory markers detected, in serum or arteries?

5.     7. and 8., both of these two section discussed the effects of oxidative stress and neuroinflammation on brain aging, the definition of neurodegenerative diseases and nervous system disorders are unclear, and some sentence are repetitive. Restructure of these two sections are recommended, which makes discussion more clear.

Author Response

First of all, thank you for thoughtful and thorough review. We have revised the manuscript based on the comments of reviewers. We would like to kindly ask you to proceed with a review. Please find below a response to comments.

This paper discussed the importance of the antioxidant effect of lycopene in inhibiting the development of such diseases as cardiovascular diseases, liver diseases, and diseases within the nervous system. An important point is that the majority of evidence of a beneficial effect of lycopene on above-mentioned diseases comes from experimental work, while the evidence from clinical studies is both less abundant and less clear on any beneficial effect. This needs to be clearly underlined in the abstract and the conclusions.

Thank you for pointing this out, we agree with the Reviewer’s viewpoint. We have made the following changes:

Lycopene is not registered as a drug, so there have not been many studies on humans so far. In the content of individual chapters, as well as in the abstract and conclusions, information was added that most of the evidence for the presence of lycopene comes from experimental studies. There is a need to conduct clinical trials to confirm the beneficial effects of lycopene in the treatment of the described diseases in humans.

 At the same time, there is only one table. On the other hand, sometimes it is really repetitive.

Thank you for pointing this out, we agree with the Reviewer’s viewpoint. A table containing studies on the effect of lycopene on diseases of the nervous system has been added to the article.

Authors should also pay attention to the points outlined below, listed in order of appearance.

  1. Line 15, please check the sentence “diseases of the etiology of etiology”

Thank you for pointing this out, we agree with the Reviewer’s viewpoint.

The abstract has been corrected. Stylistic errors have been corrected and more information has been added.

  1. “1. Lycopene”, graphic representation of lycopene, isomers, and metabolic processes in the human body are recommended.

Thank you for pointing this out, we agree with the Reviewer’s viewpoint.

Figures showing the main lycopene metabolites (Figure 1) and lycopene isomers (Figure 2) have been added.

  1. Line 156-159, the information are repetitious with above two paragraph.

Thank you for pointing this out, we agree with the Reviewer’s viewpoint.

The information in the mentioned paragraphs has been redacted and repeated sentences have been removed.

  1. Line 245-297, where were these oxidative stress related markers and inflammatory markers detected, in serum or arteries?

The publication supplemented information on where markers of oxidative stress and inflammation were detected.

  1. 7. and 8., both of these two section discussed the effects of oxidative stress and neuroinflammation on brain aging, the definition of neurodegenerative diseases and nervous system disorders are unclear, and some sentence are repetitive. Restructure of these two sections are recommended, which makes discussion more clear.

Thank you for pointing this out, we agree with the Reviewer’s viewpoint.

Chapters 7. neurodegenerative disorders and 8. disorders of the nervous system have been reworded and merged into one chapter 6 on disorders of the nervous system.

Round 2

Reviewer 1 Report

The revised manuscript looks much better now. I would like to suggest to add one more figure to indicate the molecular mechanisms of Lycopene's antioxidant effect involving in fighting diseases.

No additional comments.

Author Response

We are thankful to the Reviewer for the positive evaluation of our paper. We thank the Reviewer for this interesting comment. We have added the following  figure in revised manuscript.

Reviewer 2 Report

The manuscript has been sufficiently improved 

Author Response

We are thankful to the Reviewer for the positive evaluation of our paper.